# Tuning charge and correlation effects for a single molecule on a graphene device

Sebastian Wickenburg[1,2,*], Jiong Lu[1,3,4,*], Johannes Lischner[1,5], Hsin-Zon Tsai[1], Arash A. Omrani[1], Alexander Riss[1,6], Christoph Karrasch[1,7], Aaron Bradley[1], Han Sae Jung[1], Ramin Khajeh[1], Dillon Wong[1], Kenji Watanabe[8], Takashi Taniguchi[8], Alex Zettl[1,2,9], A.H. Castro Neto[4,10], Steven G. Louie[1,2] & Michael F. Crommie[1,2,9]

The ability to understand and control the electronic properties of individual molecules in a device environment is crucial for developing future technologies at the nanometre scale and below. Achieving this, however, requires the creation of three-terminal devices that allow single molecules to be both gated and imaged at the atomic scale. We have accomplished this by integrating a graphene field effect transistor with a scanning tunnelling microscope, thus allowing gate-controlled charging and spectroscopic interrogation of individual tetrafluoro-tetracyanoquinodimethane molecules. We observe a non-rigid shift in the molecule's lowest unoccupied molecular orbital energy (relative to the Dirac point) as a function of gate voltage due to graphene polarization effects. Our results show that electron–electron interactions play an important role in how molecular energy levels align to the graphene Dirac point, and may significantly influence charge transport through individual molecules incorporated in graphene-based nanodevices.

[1] Department of Physics, University of California, Berkeley, California 94720, USA. [2] Materials Sciences Division, Lawrence Berkeley National Laboratory, Berkeley, California 94720, USA. [3] Department of Chemistry, National University of Singapore, 3 Science Drive 3, Singapore 117543, Singapore. [4] Centre for Advanced 2D Materials and Graphene Research National University of Singapore, 6 Science Drive 2, 117546 Singapore, Singapore. [5] Department of Materials, Imperial College London, Prince Consort Rd, London SW7 2BB, UK. [6] Physik-Department E20, Technical University of Munich, 85748 Garching, Germany. [7] Dahlem Center for Complex Quantum Systems and Fachbereich Physik, Freie Universität Berlin, 14195 Berlin, Germany. [8] National Institute for Materials Science, 1-1 Namiki, Tsukuba 305-0044, Japan. [9] Kavli Energy NanoSciences Institute at the University of California Berkeley and the Lawrence Berkeley National Laboratory, Berkeley, California 94720, USA. [10] Department of Physics, National University of Singapore, 2 Science Drive 3, Singapore 117542, Singapore. * These authors contributed equally to this work. Correspondence and requests for materials should be addressed to J.L. (email: chmluj@nus.edu.sg) or to S.G.L. (email: sglouie@berkeley.edu) or to M.F.C. (email: crommie@berkeley.edu).

Creating electronic devices based on single molecules is a key goal of modern nanotechnology[1–9]. Future progress in this area, however, hinges on developing a better understanding of the fundamental properties of individual molecules in new, complex electronic environments[3–8,10–12]. Molecules have been integrated into gated three-terminal electrical devices previously, allowing continuous tuning and characterization of molecular electronic properties[2–8,10,11,13–17]. Precise interpretations, however, have been made more difficult in these experiments by the fact that local chemical structures have not been well-characterized due to an inability to image individual molecules in device junctions. Single molecules with well-characterized chemical structure, on the other hand, are regularly studied via two-terminal scanning tunnelling microscopy (STM) techniques where electronic properties are *not* tunable via a third external gate electrode[11,12,15,18–21] (as opposed to internal gating/doping generated by local impurity configurations[12,20,22]). Some progress has been made at introducing gate electrodes into scanned probe measurements of molecular systems[23], but gate-tunable control of single-molecule charge states has not yet been demonstrated.

Here we combine STM and non-contact atomic force microscopy (nc-AFM) to demonstrate gate-tunable control of the charge state of individual, well-characterized tetrafluoro-tetracyanoquinodimethane (F$_4$TCNQ) molecules at the surface of a graphene field effect transistor (FET) (F$_4$TCNQ is a commonly used acceptor in molecular electronics). This system allows the substrate Fermi energy ($E_F$) to be continuously tuned all the way through the lowest unoccupied molecular orbital (LUMO) energy of a single F$_4$TCNQ molecule. Using STM spectroscopy we have determined the gate-dependent energetic evolution of the LUMO level ($E_L$) and its associated vibronic modes relative to the graphene Dirac point ($E_D$). We show that the energy alignment between $E_L$ and $E_D$ changes as the substrate charge carrier density is tuned by gating, indicating the presence of electron–electron interactions that renormalize the molecular quasiparticle energy. This is attributed to gate-tunable image-charge screening in graphene and is corroborated by *ab initio* calculations. Our findings reveal that such tunable electronic correlation effects significantly renormalize the electron addition and removal energies for individual molecules incorporated into graphene devices.

## Results

**Topographic imaging of anchored F$_4$TCNQ molecules.** The graphene FETs used here were made by transferring graphene grown via CVD techniques onto BN flakes supported by an SiO$_2$ layer at the surface of a doped Si wafer (the doped Si provides the FET back-gate)[23,24]. Use of a BN substrate reduces charge inhomogeneity in graphene, allowing us to better probe intrinsic molecule/graphene electronic properties[24]. F$_4$TCNQ (Fig. 1a) was selected for this study because its LUMO state has been predicted to lie close to the graphene Dirac point[25], thus facilitating molecular charge state tunability. Scanning probe measurements of molecule-decorated devices were performed at $T = 5$ K in ultra-high vacuum. Figure 1b shows an STM image of individual F$_4$TCNQ molecules adsorbed onto the surface of a graphene/BN device at low coverage. F$_4$TCNQ exhibits a dog-bone-like shape that resembles the LUMO of an isolated F$_4$TCNQ molecule, similar to previous measurements of F$_4$TCNQ molecules on graphene/metal[26–29]. Individual F$_4$TCNQ molecules are not strongly pinned by the graphene/BN substrate, and so are prone to move quite easily when subjected to the local tip-induced electric fields required for high-resolution STM spectroscopy.

To overcome this problem, we devised an anchoring strategy to immobilize individual F$_4$TCNQ molecules by using electronically inert 10,12-pentacosadiynoic acid (PCDA) as a molecular anchor (Fig. 1a). We evaporated PCDA molecules onto graphene/BN just before deposition of F$_4$TCNQ molecules. Our STM images reveal that PCDA molecules self-assemble into ordered islands on graphene/BN that exhibit straight edges (Fig. 1c), consistent with previously reported behaviour of PCDA on graphene/SiC (ref. 30). Individual F$_4$TCNQ molecules anchor nicely to the edge of PCDA islands, as seen in Fig. 1c. These edge-anchored F$_4$TCNQ molecules were sufficiently stable for high-resolution STS and nc-AFM measurements. Nc-AFM with a CO tip was used to determine the precise adsorption geometry of F$_4$TCNQ on the PCDA-functionalized graphene/BN surface, as shown in Fig. 1d. In such images contrast is caused by short-range chemical forces, and bright areas exhibiting high frequency shift tend to represent surface regions with higher electron density (such as atoms and chemical bonds[31–35]). The nc-AFM image in Fig. 1d thus reveals the 'wire-frame' chemical structure and adsorption geometry of F$_4$TCNQ molecules attached to PCDA molecular anchors. Our results show that F$_4$TCNQ molecules align along the armchair direction of graphene and that there is no significant chemical interaction between F$_4$TCNQ and PCDA (the faint lines seen connecting the F$_4$TCNQ and PCDA molecules are a common feature for adsorbates bound by weak hydrogen and van der Waals interaction)[35–38].

**Electronic structure of F$_4$TCNQ on graphene.** We measured the electronic structure of individual F$_4$TCNQ molecules anchored to PCDA on our graphene/BN devices through the use of STM spectroscopy (the F$_4$TCNQ electronic structure was not significantly affected by the nearby PCDA (Supplementary Fig. 1)). The inset in Fig. 2a shows a typical d$I$/d$V$ spectrum measured over the range $-0.4$ V $< V_S < 0.4$ V ($V_S$ is the sample voltage with respect to the tip) with the back-gate held at $V_G = 0$ V. Spectroscopy was acquired on the outer edges of the molecule to avoid inelastic tunnelling effects[39] (Supplementary Fig. 2). Two broad peaks are visible in the $V_S > 0$ V range, marked $L$ and $L'$. These peaks are asymmetric, as seen in the adjacent high-resolution d$I$/d$V$ spectrum in Fig. 2a. For $V_S < 0$ V only one peak is seen, marked 'charging', which is significantly sharper than the $V_S > 0$ V peaks[21,40,41]. The $L$ peak ($V_S > 0$) is derived from the F$_4$TCNQ LUMO state. The charging peak ($V_S < 0$), on the other hand, does not directly indicate a feature in the molecular density of states but rather occurs due to tip-induced band bending as the tip electric field pulls the LUMO state below the Fermi energy ($V_S = 0$ V), causing it to fill with charge[21,40–43]. We attribute the asymmetric structure of the $L$ and $L'$ peaks to vibronic sidebands, as has been observed in other systems[23,39,44,45]. To extract the experimental energies of the vibrations involved, we fit the F$_4$TCNQ spectrum to a sum of Gaussian peaks. The $L$ feature is fit well by a peak located at $E_L = 61 \pm 6$ meV (blue dashed line in Fig. 2a) along with five other satellite peaks evenly spaced every $37 \pm 7$ meV (orange dashed lines). The $L'$ feature is similarly fit well by a peak located at $E_{L'} = 288 \pm 23$ meV (purple dashed line) with three additional satellite peaks evenly spaced every $37 \pm 7$ meV (orange dashed lines) (see Supplementary Fig. 3 for details of fit).

To test our hypothesis regarding the vibrational origin and structure of the $L$ and $L'$ peaks, we calculated the hybridized molecular orbitals and vibrational modes of an F$_4$TCNQ molecule on graphene, as well as the associated electron–phonon coupling, from first principles. As shown in Fig. 2b, numerous vibrational modes exist in the range $0 < E < 300$ meV.

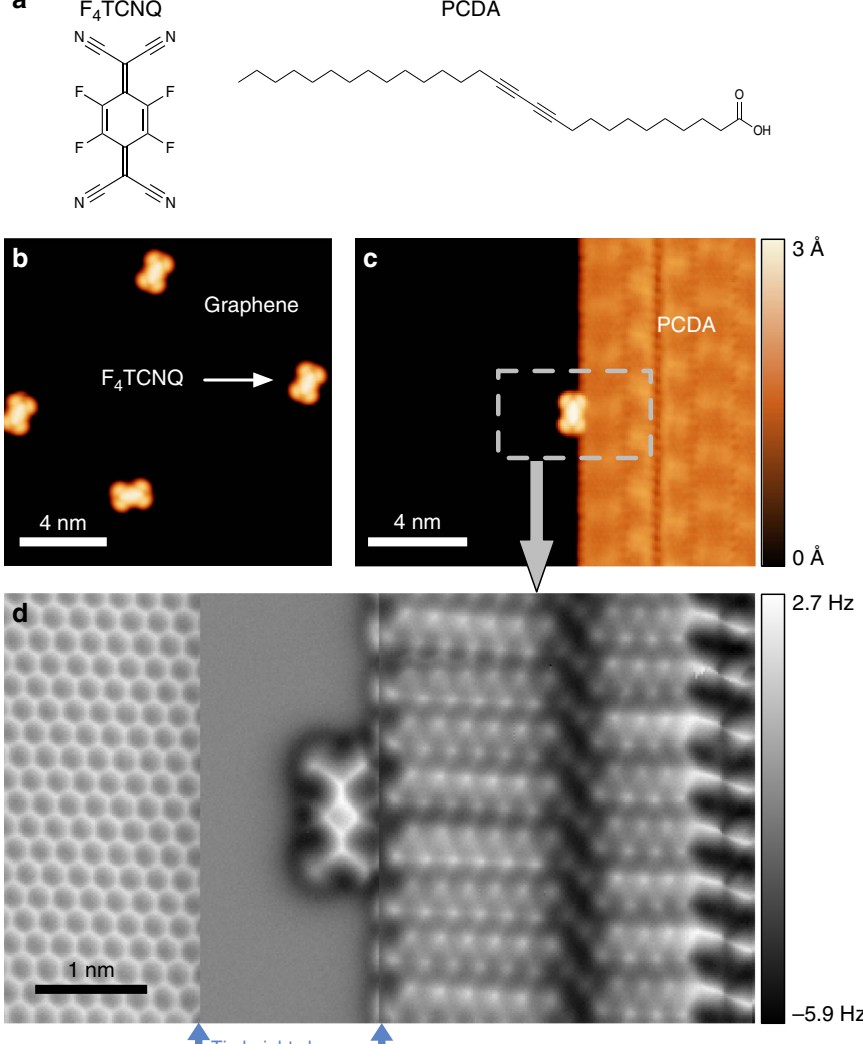

**Figure 1 | STM and nc-AFM images of $F_4$TCNQ and PCDA on graphene.** (**a**) Chemical structure of $F_4$TCNQ and PCDA molecules. (**b**) STM image of $F_4$TCNQ molecules decorating graphene/BN substrate ($V_S = 2$ V, $I_t = 5$ pA). (**c**) Deposition of PCDA followed by deposition of $F_4$TCNQ results in well-ordered PCDA islands with isolated $F_4$TCNQ molecules adsorbed at the island edges ($V_S = 2$ V, $I_t = 10$ pA). (**d**) $F_4$TCNQ molecular adsorption geometry is shown with single-chemical-bond resolution *via* nc-AFM (qPlus) with a CO-functionalized tip. The nc-AFM image was taken in constant height mode at three different heights by lowering the tip at the two positions marked by blue arrows (320 pm (left) and 70 pm (right)). Hydrogen atoms can be resolved in the PCDA molecules, as well as triple bonds. $F_4$TCNQ molecules are seen to adsorb with their nitrogen and fluorine atoms close to the terminal hydrogen atoms of PCDA, indicating hydrogen bonding as a likely source of $F_4$TCNQ stabilization. The honeycomb lattice of graphene is clearly resolved. (All images taken at $T = 5$ K).

Calculation of the electron-phonon coupling for each of these modes, however, (Fig. 2b, green curve) shows that the strongest coupling occurs at $\omega_1 \approx 34$ meV and $\omega_2 \approx 183$ meV (Supplementary Note 1). These energies correspond to a uniaxial stretching mode with $A_g$ symmetry ($\omega_1 = 34$ meV) and a breathing mode of the inner carbon ring also having $A_g$ symmetry ($\omega_2 = 183$ meV). The calculated energies agree reasonably well with the experimental energy spacing of the peaks extracted from within the $L$ and $L'$ features of Fig. 2a ($37 \pm 7$ meV), as well as the energy difference $E_{L'} - E_L = 227 \pm 24$ meV. This shows that $L'$ is a vibronic satellite of $L$ due to phonons having energy $\omega_2$ and that the internal structure of both $L$ and $L'$ represents vibronic satellites due to phonons having energy $\omega_1$. A more detailed analysis involving a cumulant expansion to calculate the spectral function including vibronic modes also agrees with our measured spectra, supporting the vibronic interpretation of the spectral lineshape (Supplementary Note 1).

**Reversible charge state switching of molecules using a back-gate.** A unique aspect of this study is that we are able to reversibly control the charge state of a single $F_4$TCNQ molecule by continuously tuning the substrate Fermi level past the LUMO energy level via application of an electrostatic back-gate. Figure 2c shows STM d$I$/d$V$ spectra of a single anchored $F_4$TCNQ molecule at two different back-gate voltages ($V_G$). Here we label the empty and occupied LUMO orbital as LUMO$^0$ and LUMO$^-$, respectively. The blue curve acquired at $V_G = -50$ V shows the LUMO level well above $E_F$ and thus empty (LUMO$^0$), resulting in a neutral molecule at this gate voltage. In contrast, the red d$I$/d$V$ trace acquired at $V_G = +30$ V shows the LUMO well below $E_F$ and thus filled by an electron, causing the molecule to become negatively charged (LUMO$^-$). A notable difference between the red and blue curves is that the vibrational sidebands for the LUMO$^-$ state extend downwards to more negative $V_S$ values compared with the sidebands for LUMO$^0$ which extend upwards

to more positive $V_S$ values. This can be explained by the fact that higher-energy electron-like vibronic excitations (for $LUMO^0$) occur at higher values of $V_S$, whereas higher-energy hole-like vibronic excitations (for $LUMO^-$) occur at lower values of $V_S$. Our first-principles calculation of the spectral function using a cumulant expansion[46] confirms this intuitive electron–hole symmetry and reproduces the observed vibronic spectra (Supplementary Fig. 4).

**Non-rigid shift of LUMO energy.** Our ability to gate the substrate of an adsorbed molecule allows us to address the

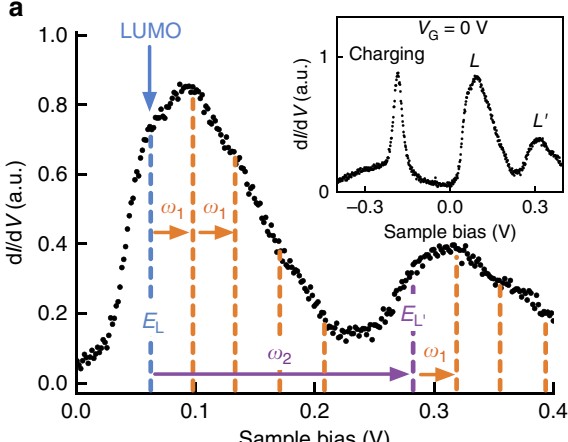

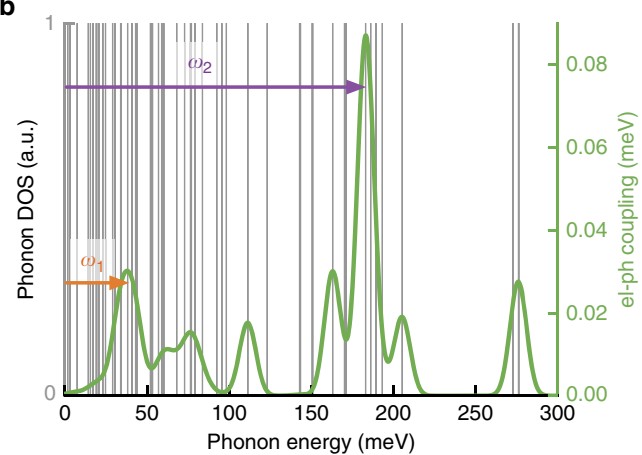

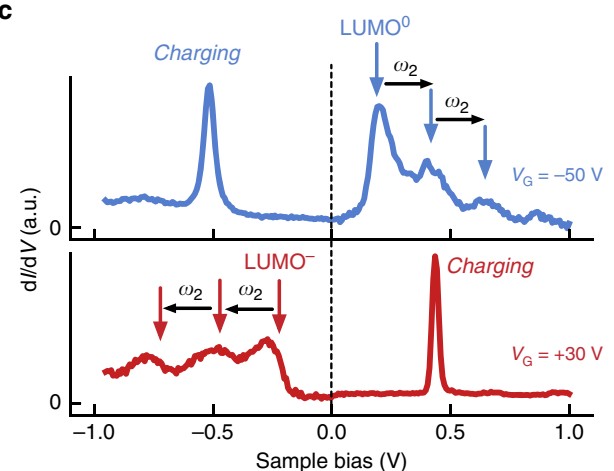

fundamental issue of the importance of electron–electron interactions for molecule/substrate systems. We do this by measuring how the energy level of the molecular orbital changes relative to the graphene band structure as $E_F$ is swept by the gate. In the absence of electron–electron interactions we expect the position of the molecular level relative to $E_D$ to be independent of $E_F$. If, on the other hand, electron–electron interactions play a significant role then we expect the LUMO energy ($E_L$) to shift relative to $E_D$ as $E_F$ is tuned. We measured this effect by acquiring $dI/dV$ spectra both on a single $F_4$TCNQ molecule (to obtain $E_L$) and off the molecule (to obtain $E_D$) as a function of gate voltage. Figure 3a shows a plot of the resulting 'on molecule' $dI/dV$ spectra as a function of gate voltage ($-40\,V < V_G < +40\,V$), while Fig. 3b shows a plot of the 'off molecule' spectra (that is, for bare graphene). The on-molecule spectra show the molecular LUMO level continuously sweeping from an empty orbital state ($LUMO^0$) for $V_G < 0\,V$ to a filled orbital state ($LUMO^-$) for $V_G > 0\,V$. Similarly, the off-molecule spectra show the graphene Dirac point sweeping from above $E_F$ for $V_G < 0\,V$ (the $p$-doped regime) to below $E_F$ for $V_G > 0\,V$ (the $n$-doped regime). To extract the experimental gate-dependent values of $E_L$, we fit the $dI/dV$ spectra in Fig. 3a with sums of Gaussians. The resulting LUMO energies are shown in Fig. 3a at the red dot locations. The gate-dependent $E_D$ values were obtained from the spectra of Fig. 3b by fitting inverted Gaussians to the prominent local minimum of each spectrum (resulting in the black dots shown in Fig. 3b).

Figure 4 shows a direct comparison of the experimental $E_D$ and $E_L$ values as a function of gate voltage (because $E_D$ is difficult to obtain for some gate voltages we use values here that are obtained by fitting a characteristic square root function to the data of Fig. 3b (Supplementary Fig. 5)). For small gate voltage (low charge carrier density) $E_D$ and $E_L$ are seen to lie almost directly on top of each other. When the gate voltage magnitude is increased (resulting in higher charge carrier density) however, the values diverge, separating by as much as 100 meV at the highest gate voltage (corresponding to a charge carrier density of $\sim 3 \times 10^{12}\,cm^{-2}$ at $V_G = 60\,V$). The energy difference between $E_D$ and $E_L$ ($E_D$–$E_L$) is seen to increase monotonically with increasing charge carrier density for both electrons and holes, suggesting that electron–electron interactions play a role in determining $E_L$ for this adsorbate system. The observation that $E_D$–$E_L$ does not depend on the polarity of graphene charge carriers rules out simple band bending as an explanation for the energy shift, since band bending would shift $E_L$ to higher energies when $E_L$–$E_F > 0$, opposite to what is observed[21,40,47].

**Figure 2 | STS spectra of F$_4$TCNQ molecules reveal tunable vibronic modes.** (**a**) $dI/dV$ spectrum for a single F$_4$TCNQ molecule on graphene/BN shows two main peaks spaced by $\sim 227$ meV for $V_S > 0\,V$, and one peak for $V_S < 0\,V$. Peaks for $V_S > 0\,V$ originate from LUMO and vibronic modes while the peak at $V_S < 0\,V$ originates from tip-induced charging of the LUMO level. Initial tunnelling parameters: $I_t = 30$ pA , $V_S = 0.4$ V, $V_{AC} = 8$ mV. (**b**) *Ab initio* calculated energies of phonon modes for F$_4$TCNQ/graphene (grey), as well as electron-phonon coupling strength between phonon modes and LUMO state (green curve, broadened by a 12 meV full-width Gaussian). The phonons with highest electron-phonon coupling occur at $\omega_1 \sim 34$ meV and $\omega_2 \sim 183$ meV. (**c**) $dI/dV$ spectrum of F$_4$TCNQ/graphene/ BN for $V_G = -50$ V (blue) shows F$_4$TCNQ vibronic states for a neutral molecule ($LUMO^0$). $dI/dV$ spectrum of the same molecule at $V_G = 30$ V (red) shows that vibronic states for a charged molecule ($LUMO^-$) switch their energy alignment from increasing energy ordering to decreasing energy ordering when the charge state is switched by the gate (the portion of the blue (red) curve below (above) $E_F$ has been scaled by 0.4 (0.2) to fit on the plot). Initial tunneling parameters: $I_t = 15$ pA , $V_S = 1$ V, $V_{AC} = 12$ mV.

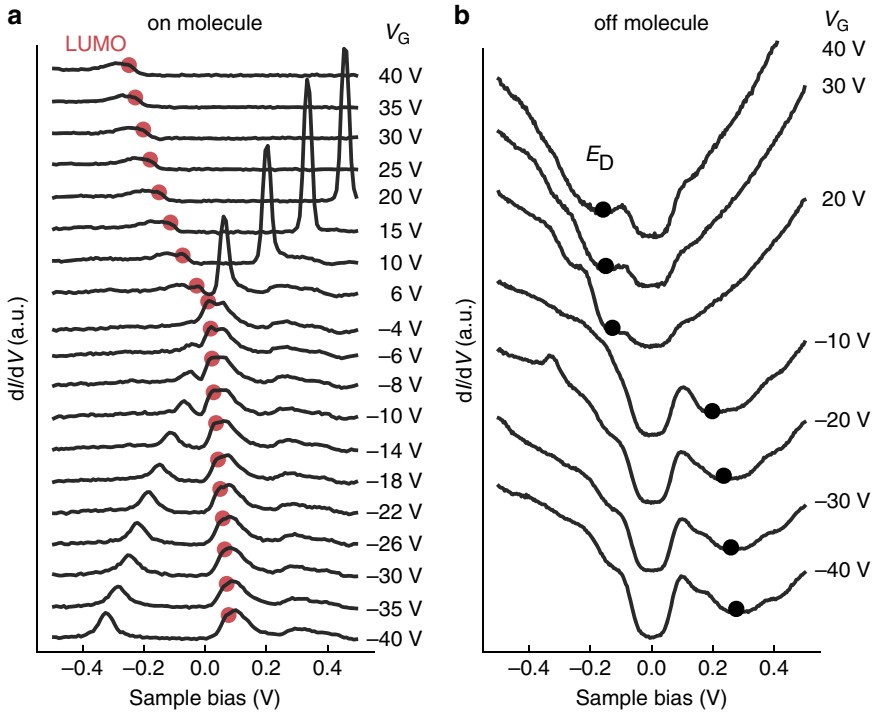

**Figure 3 | Gate-dependent STS of individual F$_4$TCNQ molecule on graphene/BN.** (**a**) d$I$/d$V$ spectra obtained with STM tip held over a single F$_4$TCNQ molecule recorded at different gate voltages show that the LUMO state and vibronic sidebands shift relative to $E_F$ as the gate is swept. Red dots mark energy locations of the LUMO state at different gate voltages, extracted by fitting a sum of Gaussian peaks to the d$I$/d$V$ spectra (initial tunneling parameters: $I_t = 15$ pA , $V_S = 1$ V, $V_{AC} = 12$ mV). (**b**) d$I$/d$V$ spectra obtained with the STM tip held over a bare patch of graphene/BN near an F$_4$TCNQ molecule (distance = 4 nm) recorded at different gate voltages show dependence of Dirac point energy ($E_D$) on gate voltage. Black dots mark Dirac point obtained by fitting inverted Gaussians to the minimum of each spectrum. Final $E_D$ values are obtained by subtracting the inelastic phonon energy of 63 meV from these measured features[54,55] (initial tunneling parameters: $I_t = 60$ pA, $V_S = 0.5$ V, $V_{AC} = 12$ mV).

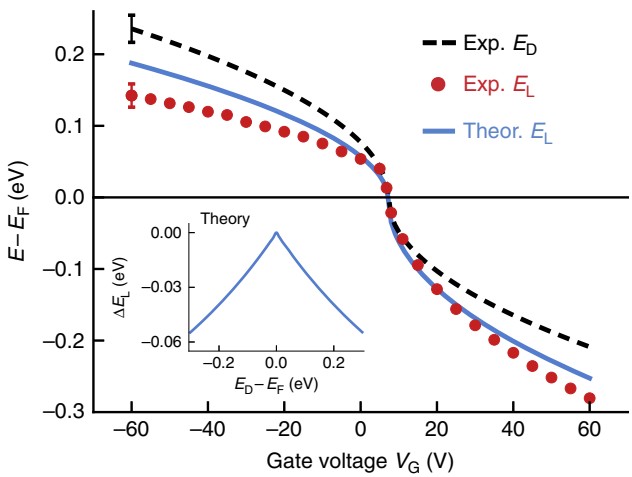

**Figure 4 | LUMO and Dirac point energies as a function of gate voltage.** The experimental LUMO energies (red dots) and the theoretically predicted ones (blue curve) agree qualitatively by lying below the Dirac point energy (dashed line, see Supplementary Fig. 5 for fitting details) for all gate voltages away from charge neutrality. The experimental error in the LUMO energy is estimated as the mean half-width of the Gaussian fits to the F$_4$TCNQ spectra (16 meV). The experimental error in $E_D$ is estimated using the fit of the measured $E_D$ to the square root dependence of $E_D$ versus $V_G$ (the r.m.s. of the fit residuals = 19 meV). Inset: theoretical energy renormalization of the LUMO level due to interaction of LUMO charge with induced graphene image charge as a function of $E_F$.

## Discussion

We are able to explain this behaviour as the result of Coulomb interaction between charge added to an F$_4$TCNQ molecular orbital (*via* tunnelling), and the electronic polarization that it induces in the graphene substrate. This many-electron interaction always lowers the energy of a system, since the interaction between charge added to an adsorbate and its image charge in the substrate is always attractive. Such effects are well known to reduce the energy gap between affinity and ionization levels for adsorbates on conventional substrates[13,48–50]. For an adsorbate on graphene this effect is expected to be tunable since polarizability depends on the density of states at $E_F$, which is readily changed by gating a graphene device.

To estimate the expected magnitude of this effect for comparison to our data, we calculated how the F$_4$TCNQ LUMO energy level is renormalized via screening within many-body perturbation theory (standard DFT treatments of the Kohn–Sham eigenvalues do not take this non-local effect into account). This was accomplished by modelling an electron that has tunnelled into the empty LUMO level as a point charge located a distance $z^*$ above the graphene plane ($z^*$ is estimated to be 3 Å from our *ab initio* calculations of F$_4$TCNQ/graphene, in reasonable agreement with the tip-height change during AFM measurements (Fig. 1d)). The point charge exposes the graphene substrate to a Coulomb potential, $\phi_{ext}(\mathbf{r})$, which induces screening charge density, $n_{ind}(\mathbf{r})$, in the graphene. We calculated $n_{ind}(\mathbf{r})$ and the change it causes to the electrostatic potential, $\phi_{ind}(\mathbf{r})$, within linear response theory using the RPA dielectric function of graphene[53] (the BN substrate was taken into account by choosing a background dielectric constant of $\epsilon_{bg} = 2$,

see Supplementary Notes 2 and 3). The screening-induced lowering of the LUMO energy, $\Delta E_L$, was estimated by evaluating $\phi_{ind}(\mathbf{r})$ at the location of the point charge above the graphene surface thus yielding $\Delta E_L = -\frac{1}{2}e\phi_{ind}(z^*)$ (ref. 48). The inset in Fig. 4 shows the calculated $\Delta E_L$ values as a function of $E_F$. For $E_F = E_D$ the energy correction is smallest since graphene has no density of states at the Fermi level when $E_F$ is aligned to the Dirac point. As $E_F$ shifts away from $E_D$, the energy correction increases equally for both electron and hole-doping since the carrier RPA dielectric function for graphene is electron–hole symmetric. The near-linear dependence of $\Delta E_L$ on $E_F$ stems from the linear graphene density of states, $\rho(E)$, and the fact that the graphene electronic susceptibility is proportional to $\rho(E_F)$.

To compare the calculated $\Delta E_L$ to spectroscopy taken on individual $F_4TCNQ$ molecules, we can add it to our gate-dependent measurements of $E_D$ (taking into account a small offset constant observed experimentally at charge neutrality). In the absence of any screening effects ($\Delta E_L = 0\,eV$), this would result in $E_L \approx E_D$ at every gate voltage (the simple 'rigid shift' case). The Coulomb-induced renormalization effects described above, however, cause this procedure to result in a nontrivial shift of $E_L$ with respect to $E_D$ as a function of gate voltage. Figure 4 shows a plot of the resulting renormalized $E_L$ value (blue curve) as a function of gate voltage compared with the experimental $E_L$ values (red dots). For small gate voltages the renormalized $E_L$ value coincides with $E_D$, but as the gate voltage (and carrier density) magnitude is raised the calculated $E_L$ values fall increasingly downward compared with $E_D$, just as seen for the experimental $E_L$ values. The model thus captures both the observed independence of the LUMO renormalization on the graphene carrier type (electrons or holes) and also reproduces the general magnitude of the molecular orbital energy change. The match is not perfect (the experimental drop in $E_L$ tends to be larger than the calculated $\Delta E_L$, but is reasonable considering that the calculation has no adjustable parameters. One possible source for the discrepancy between theory and experiment are intra-molecular electron–electron interactions, which are not accounted for in the simple image charge model. We have examined the effect of these interactions on the LUMO energy using an Anderson model approach and find that they do lead to additional LUMO energy renormalization, but smaller in magnitude than the image charge effects (Supplementary Note 4).

In conclusion, we demonstrate reversible tuning of the charge state of individual $F_4TCNQ$ molecules using an electrostatically back-gated graphene device. The molecular adsorption geometry is imaged via nc-AFM with single-chemical-bond resolution and the gate-dependent molecular electronic structure is determined via STM spectroscopy. We find that molecular vibronic modes can be switched from electron-like energy alignment to hole-like energy alignment depending on the tunable molecular charge state, in agreement with cumulant-expansion theory. We additionally observe a non-rigid shift in the LUMO energy relative to the Dirac point as a function of gate voltage that can be explained by many-electron interaction renormalization of the LUMO energy caused by tunable substrate polarization effects.

## Methods

**Graphene device fabrication.** A back-gated graphene/BN/SiO$_2$ device was prepared by overlaying CVD-grown graphene onto hexagonal boron nitride (h-BN) flakes exfoliated onto a SiO$_2$/Si substrate. h-BN flakes were exfoliated onto heavily doped silicon wafers and annealed at 500 °C for several hours in air prior to graphene transfer. The graphene was grown on copper foil by the CVD method[51] and transferred to the BN/SiO$_2$ substrate via a poly methyl methacrylate stamp[52]. Electrical contact was made to the graphene by depositing Ti (10 nm thick)/Au (30 nm thick) electrodes using the stencil mask technique.

**Sample preparation.** The graphene device was first annealed in flowing Ar/H$_2$ gas at 350 °C and then annealed subsequently in UHV at $T \sim 350$ °C for several hours

until an atomically clean surface was achieved before the deposition of molecules. PCDA and $F_4TCNQ$ were deposited consecutively onto the clean graphene substrate at room temperature using Knudsen cell evaporators in the UHV chamber.

**STM/AFM measurements.** STM/nc-AFM measurements were performed using a qPlus-equipped commercial Omicron LT-STM/AFM under UHV conditions at $T = 5\,K$ using tungsten tips. STM topography was obtained in constant-current mode. STM tips were calibrated on a Au(111) surface by measuring the Au(111) Shockley surface state before all STS measurements. STS was performed under open feedback conditions by lock-in detection of an alternating current tunnel current with a bias modulation of 6–16 mV (r.m.s.) at 400 Hz added to the tunnelling bias. The tips were functionalized for nc-AFM imaging by picking up individual CO molecules on a Au(111) surface[33]. nc-AFM images were recorded by measuring the frequency shift of a qPlus resonator, while scanning over the molecule in constant-height mode ($f_0 = 28.7$ kHz, $Q = 90$ k, $A = 60$ pm). WSxM software was used to process all STM/nc-AFM images[53].

**Data availability.** The data that support the findings of this study are available from the corresponding author on reasonable request.

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

## Acknowledgements

This research was supported by the Director, Office of Science, Office of Basic Energy Sciences, Materials Sciences and Engineering Division, of the US Department of Energy under contract no. DE-AC02-05CH11231 (Nanomachine program-KC1203) (STM imaging and spectroscopy, cumulant-expansion studies of spectral line shapes and phonon sidebands), Molecular Foundry (graphene growth, growth characterization), National Science Foundation grant DMR-1206512 (sample fabrication) and National Science Foundation grant DRM-1508412 (electron-phonon coupling calculations). K.W. and T.T. acknowledge support from the Elemental Strategy Initiative conducted by the MEXT, Japan and a Grant-in-Aid for Scientific Research on Innovative Areas 'Science of Atomic Layers' from JSPS. J. Lu and A.H.C.N. acknowledge the support from the National Research Foundation, Prime Minister Office, Singapore, under its Medium Sized Centre Programme and CRP award no. R-144-000-295-281. J. Lischner acknowledges support from EPSRC under grant no. EP/N005244/1 (development of image charge model). Computational resources have been provided by DOE at NERSC. A.R. acknowledges fellowship support by the Austrian Science Fund (FWF): J3026-N16.

## Author contributions

S.W., J. Lu, H.-Z.T. and A.O. designed and performed experiments, analysed data and wrote the paper. J. Lischner conceived of and performed theoretical calculations, and wrote the paper. C.K. performed Anderson model calculations and gave conceptual advice. A.R. designed the experiments, gave technical support and conceptual advice, and edited the manuscript. A.B., D.W. gave technical support and conceptual advice and helped with the experiments. H.S.J. and R.K. facilitated the sample fabrication and gave technical support. K.W. and T.T. gave technical support and grew h-BN for the device. S.G.L supervised the theoretical calculations. S.G.L, A.H.C.N., A.Z. and M.F.C. coordinated the collaboration. M.F.C. designed and supervised the experiments, supervised the data analysis and wrote the paper.

## Additional information

**Competing financial interests**: The authors declare no competing financial interests.

**How to cite this article**: Wickenburg, S. *et al.* Tuning charge and correlation effects for a single molecule on a graphene device. *Nat. Commun.* **7,** 13553 doi: 10.1038/ncomms13553 (2016).

**Publisher's note**: 

