## [Peer Review File · Nature Communications]

REVIEWERS' COMMENTS:

Reviewer #1 (Remarks to the Author):

Manuscript #: NCOMMS-16-18189-T

Title: Tuning Charge and Correlation Effects for a Single Molecule on a Graphene Device.

In this manuscript the authors probed the reversible tuning of a charge state in an isolated molecule on a back-gated graphene device. This sophisticated experiment utilized advanced graphene device development, molecular deposition, scanning probe techniques, and computational theory and modeling. The result is the tuning of the charge state of F4TCNQ molecules corresponding to the shift in Dirac point of the graphene device. Molecular vibronic modes were modeled and characterized with experimental shifting of this mode from electron to hole energy alignment.

Overall, this is an outstanding experiment that is at the cutting-edge of molecular technology interfaced with low-dimensional devices. It is no surprise that to successfully implement the high quality results and analysis at every level (device fab, theory, experimental implementation) involved a collaboration of top leaders within the field (reflected in the authorship). The experimental data is beautiful, while coupled with theory and modeling provide a compelling interpretation of the results.

I found almost no flaws with the manuscript and feel that this is an excellent contribution for Nature Communications and its readership. I encourage the Editor to publish the manuscript as is.

If I had any recommendations for this manuscript it would be to be more clear for the reader as to why I should care about F4TCNQ ... why is this molecule important.

My final comments to the authors would be to challenge them to think about: Now that you are pulling off these technically challenging experiments is the final goal achieved, is the desire to resurrect molecular electronics, or is this the platform for new breakthrough technology?

Reviewer #3 (Remarks to the Author):

This paper reports gating of a molecule adsorbed on graphene by means of a gate/insulator stack underneath the graphene. Gating is detected by scanning tunneling spectroscopy of the F4-TCNQ molecule. This experiment is elegantly designed and the analysis of the STS spectra has been done carefully with the help of theory. A particular point that the authors stress is that the LUMO level shifts in response to the gate voltage, but it does not shift the same amount (energy) as the Dirac point of graphene. The shift the TCNQ molecule occurs because of electron correlation effects between the graphene and the TCNQ. That is, it appears that the authors do not ascribe the shifting the molecule LUMO to the "field effect", but rather the field effect is envisioned as tuning the electron density in graphene and then subsequent coupling of the graphene to the TCNQ results in the shift of the TCNQ LUMO.

The paper is exciting in many respects as it offers an alternative approach to molecular gating. The experimental and theoretical results are carefully presented; the paper is of high quality.

I have the following comments:

1. It would be very useful to have a very simple energy level diagram showing the Dirac point of graphene and the LUMO of the F4 TCNQ under different bias conditions.

2. Nowhere in the paper is the phrase "field effect" employed. This is no doubt intentional, but it would be useful for the authors to compare their mechanism of molecular gating to the mechanism presumed to work in a lateral break junction.

3. What is the assignment for L'?

DEPARTMENT OF PHYSICS
366 Le Conte Hall #7300
TEL: 510/642-7166
FAX: 510/643-8497

10/13/2016

Reviewer #1 (Remarks to the Author):

Manuscript #: NCOMMS-16-18189-T

Title: Tuning Charge and Correlation Effects for a Single Molecule on a Graphene Device.

In this manuscript the authors probed the reversible tuning of a charge state in an isolated molecule on a back-gated graphene device. This sophisticated experiment utilized advanced graphene device development, molecular deposition, scanning probe techniques, and computational theory and modeling. The result is the tuning of the charge state of F4TCNQ molecules corresponding to the shift in Dirac point of the graphene device. Molecular vibronic modes were modeled and characterized with experimental shifting of this mode from electron to hole energy alignment.

Overall, this is an outstanding experiment that is at the cutting-edge of molecular technology interfaced with low-dimensional devices. It is no surprise that to successfully implement the high quality results and analysis at every level (device fab, theory, experimental implementation) involved a collaboration of top leaders within the field (reflected in the authorship). The experimental data is beautiful, while coupled with theory and modeling provide a compelling interpretation of the results.

I found almost no flaws with the manuscript and feel that this is an excellent contribution for Nature Communications and its readership. I encourage the Editor to publish the manuscript as is.

If I had any recommendations for this manuscript it would be to be more clear for the reader as to why I should care about F4TCNQ ... why is this molecule important.

Author response: We have added the following text to the main manuscript introduction (p.3): F4TCNQ is a commonly used acceptor in molecular electronics.

Reviewer #1: *My final comments to the authors would be to challenge them to think about: Now that you are pulling off these technically challenging experiments is the final goal achieved, is the desire to resurrect molecular electronics, or is this the platform for new breakthrough technology?*

DEPARTMENT OF PHYSICS
366 Le Conte Hall #7300
TEL: 510/642-7166
FAX: 510/643-8497

Author response: We believe that our findings show that any new breakthrough technology that incorporates molecules and graphene devices needs to take into account electron-electron interaction to optimize energy level alignment.

Reviewer #3 (Remarks to the Author):

This paper reports gating of a molecule adsorbed on graphene by means of a gate/insulator stack underneath the graphene. Gating is detected by scanning tunneling spectroscopy of the F4-TCNQ molecule. The experiment is elegantly designed and the analysis of the STS spectra has been done carefully with the help of theory. A particular point that the authors stress is that the LUMO level shifts in response to the gate voltage, but it does not shift the same amount (energy) as the Dirac point of graphene. The shift of the TCNQ molecule occurs because of electron correlation effects between the graphene and the TCNQ. That is, it appears that the authors do not ascribe the shifting of the molecule LUMO to the "field effect", but rather the field effect is envisioned as tuning the electron density in graphene and then subsequent coupling of the graphene to the TCNQ results in the shift of the TCNQ LUMO.

The paper is exciting in many respects as it offers an alternative approach to molecular gating. The experimental and theoretical results are carefully presented; the paper is of high quality.

I have the following comments:

1. It would be very useful to have a very simple energy level diagram showing the Dirac point of graphene and the LUMO of the F4 TCNQ under different bias conditions.

Author response: We believe the energy alignment of molecule LUMO and graphene Dirac point is contained in Figure 4 of the main text which shows both the experimental and theoretically predicted LUMO energies, as well as the graphene Dirac point energy for a number of different gate voltage conditions.

Reviewer #3: *2. Nowhere in the paper is the phrase "field effect" employed. This is no doubt intentional, but it would be useful for the authors to compare their mechanism of molecular gating to the mechanism presumed to work in a lateral break junction.*

Author response: We use the words "field effect" when describing our device as a "graphene field effect transistor" in the abstract and in the introduction. Our molecular gating mechanism is based on the field effect, however it is strongly modified by electron-electron interactions.

Reviewer #3: *3. What is the assignment for L'?*

Author response: We have added the following text to the main manuscript (p.6) to clarify the assignment of L':

DEPARTMENT OF PHYSICS
366 Le Conte Hall #7300
TEL: 510/642-7166
FAX: 510/643-8497

“This shows that L' is a vibronic satellite of L due to phonons having energy ω_2 and that the internal structure of both L and L' represents vibronic satellites due to phonons having energy ω_1 .”

Sincerely,

Michael F. Crommie
Professor of Physics
U.C. Berkeley Physics Department
Berkeley, CA 94720
crommie@berkeley.edu
(510) 642-939